# Measurement of Tourism-Related CO_2_ Emission and the Factors Influencing Low-Carbon Behavior of Tourists: Evidence from Protected Areas in China

**DOI:** 10.3390/ijerph20021277

**Published:** 2023-01-10

**Authors:** Jing Wu, Shen Wang, Yuling Liu, Xuesong Xie, Siyi Wang, Lianhong Lv, Hong Luo

**Affiliations:** 1Chinese Research Academy of Environmental Sciences, Beijing 100012, China; 2School of Economics and Management, Beijing Forestry University, Beijing 100083, China

**Keywords:** carbon dioxide emissions, tourist behavior, Qinling Mountains, life cycle assessment approach

## Abstract

In the fight against climate change, future policy directions in the transition toward a green travel- and tourism-based economy include improving tourism-derived CO_2_ emission levels and guiding individual low-carbon behavior. In China, people tend to engage in outdoor adventure travel and cultural tourism in natural areas. However, limited information is available on the empirical evaluation of energy use and the CO_2_ emissions associated with tourism in protected areas. The present study used a life cycle assessment to explore energy use and CO_2_ emissions due to tourism and identify the factors driving low-carbon behavior. To these ends, survey data for the protected areas of the Qinling Mountains from 2014 to 2019 were used. The results showed that energy use and CO_2_ emissions in various tourism sectors steadily increased from 2014 to 2019, primarily because of an increase in transportation activity. This study used data derived from the calculation of CO_2_ emissions per tourist per trip to identify the various factors jointly contributing to the low-carbon behavior of tourists. These included a low-carbon attitude, low-carbon knowledge, environmental education, and policy reward. The broader implications of this study are that several emission reduction policy options are available to address the challenges inherent in sustainable tourism development and that these policies may be selected according to specific conditions. The low-carbon transformation of recreational facilities at travel destinations, policy rewards, and environmental education can regulate tourist behavior, holding the key to sustainable tourism development in protected areas.

## 1. Introduction

In the post-COVID-19 era, a common global objective is to hasten green development and lifestyle for the purposes of preserving and beautifying the planet. Therefore, many nations are implementing measures and programs to promote carbon neutrality. The green and low-carbon sustainable development of tourism plays an important role in attaining these goals. Tourism has been identified as an important contributor to global anthropogenic greenhouse gas (GHG) emissions, accounting for about 5% of all global CO_2_ emissions in 2005 when tourist transportation, accommodation, and recreation activity are taken into account [1]. CO_2_ emissions from tourism (excluding aviation) might grow at a rate of 2.5% per year until 2035, even when technological and managerial advances and progress are considered [2]. According to the latest report from the World Tourism Organization, CO_2_ emissions from the tourism sector are expected to increase by ≥25%, as of 2030 [3]. While the COVID-19 pandemic led to a 7% reduction in global GHG emissions, it is expected that GHG emissions from tourism could quickly rebound as tourism operations resume [4]. Therefore, it is crucial to transform tourism operations so that this commercial sector remains in line with international climate protection goals [3]. In the fight against climate change, future policy directions in the transition toward a green travel- and tourism-based economy include improving tourism-derived CO_2_ emission levels and guiding individual low-carbon behavior.

Since the early 1990s, nature-based tourism in China has expanded to ~80% of all protected areas nationwide [5]. By the end of 2019, 2750 protected areas covering 15% of the total land area were established. These included ten pilot areas in the national park system and 474 national nature reserves. In the tourism context, protected areas have emerged as an important concept in recent outdoor recreation research [6]. In protected areas, tourism activity is usually restricted to those areas where the environmental impact can be minimized, and the protected areas are often clearly delineated. As popular tourist destinations for leisure activities, protected areas help to promote tourism, thereby supporting poverty alleviation and sustainable development [7]. However, the sheer rate of tourism expansion and the vulnerability of the habitats in protected areas necessitate further consideration of the potential environmental impact of tourist activity. Resource depletion and CO_2_ emissions from nonrenewable energy overconsumption challenge environmental protection in these areas. In the case of leisure travel, a vacation is a fixed product that can only be experienced and consumed at the point of production, rather than at the point of sale [8]. Hence, the negative environmental effects of tourism and the rapidly increasing number of tourists in the protected areas of China merit serious consideration. Relevant research subjects include the tourists’ ecological footprint, marginal CO_2_ abatement costs, and responsible touristic behavior in protected areas. With the ongoing growth of tourism and rising concerns about its sustainability, it will become ever more important to understand the energy requirements and carbon emissions of tourism in protected areas and to guide tourists toward low-carbon behavior. In terms of the protected areas of China, however, CO_2_ emissions and the factors influencing tourists’ low-carbon behavior have seldom been evaluated or investigated. For the sustainable development of tourism in protected areas, policies designed to mitigate the impacts of human behavior require accurate measurement of the CO_2_ emissions associated with tourism activity [9].

Several studies have highlighted the potential impact of tourism in current and future emission scenarios and have recommended the implementation of national tourism-specific mitigation policies. Nevertheless, the distribution of carbon emissions must be elucidated to design such policies. The characteristics of the tourism industry determine that there are many energy-consumption sources and CO_2_ emissions channels in the tourism sector [10]. Several studies have been conducted to explore the CO_2_ emissions associated with tourism [11,12]. Other studies measured energy use and CO_2_ emissions related to individual tourist destinations, based on quantitative assessments [13]. Tourist travel behavior has an environmental impact. Tourist-related activities, such as catering, accommodation, travel, and recreation all consume energy and produce carbon emissions. The energy expended through tourism varies according to tourist behavior and other factors [9]. CO_2_ emissions vary with travel choice, mode of transportation, and total travel distance [14]. Therefore, individual low-carbon behavior can help to reduce carbon emissions and improve environmental health [15]. Previous studies have shown how attitudes, intentions, values, and norms influence the low-carbon behavior of tourists [16,17,18]. However, little research has been conducted to evaluate these associations empirically within the context of tourism development in the protected areas of China. There have been numerous investigations on pro-environmental and/or low-carbon behavior, based on the theory of planned behavior (TPB). In contrast, actual low-carbon behavior has seldom (if ever) been quantified.

Against this backdrop, this paper applied the life cycle assessment (LCA) approach to calculate energy use and CO_2_ emission per tourist per trip, and further explored low-carbon behavior and its impact on individual CO_2_ emissions in the protected areas of the Qingling Mountains of China (Figure 1). The objectives of this paper were to determine the carbon emission factors, quantify the carbon emissions, and examine other factors affecting individual low-carbon behavior, to provide theoretical and practical support for the implementation of low-carbon tourism policies in the protected areas of China.

To the best of our knowledge, the present study is the first to use real-world measurements to calculate the CO_2_ emissions from tourism and identify the factors influencing the low-carbon behavior of tourists. We believe that our analyses contribute to the literature in the field in three key ways. First, few prior studies have comprehensively associated CO_2_ emissions with individual tourist activity. The present study systematically analyzed energy use and CO_2_ emissions related to tourism within China and accounted for accommodation and transportation to, within, and from recreational areas, along with the particular recreational activity. Second, prior research empirically explored the attitude–behavior gap in sustainable tourism but did not measure low-carbon behavior based on the calculated CO_2_ emissions associated with individual tourists. Consumers have demonstrated a willingness to reduce their travel-related CO_2_ emissions but have not necessarily done so in actual practice [16]. Therefore, measurements of the low-carbon behavior of tourists may be biased when these measurements are based on self-reported questions. By contrast, the present study used individual CO_2_ emission data, based on the LCA approach, to acquire authentic information regarding low-carbon consumer decisions. Third, the present study was based on a survey of 941 tourists from three nature reserves and four national forest parks in and around the protected areas of the Qinling Mountains, conducted between 2014 and 2019. These data have been drawn upon for a comparative trend analysis of tourism-driven CO_2_ emissions in the protected areas of China.

## 2. Literature Review

### 2.1. Energy Use and CO_2_ Emissions from Tourism

Previous studies on energy use and CO_2_ emissions divided the tourism industry into transportation, accommodation, and tourism activity [12]. Various studies [19,20] have indicated that transportation accounts for a significant proportion of the energy consumption and CO_2_ emissions associated with the tourism industry; more specifically, the transportation modes and the distances traveled during the journeys. Gössling et al. [19] stated that transportation accounts for most of the tourism-related CO_2_ emissions. However, these emissions can be reduced by increasing the transportation load factors. Accommodation supply and demand are vital to tourism as tourists autonomously select accommodation that offers various facilities and service levels [21]. The accommodation sector may be responsible for generating up to ~20% of all tourism-related CO_2_ emissions [1]. The intensity of energy consumption by hotels was evaluated by probing the energy utilization in various buildings and air-conditioning designs. It was also assessed by calculating the number of tourists in MJ/person–night or kg-CO_2_/person–night units. These metrics reflect the average energy consumption by one person per night when staying at a hotel [22]. According to the US Environmental Protection Agency, leisure and recreational activity accounted for ~4% of all total carbon dioxide equivalents in 2000. Kuo et al. [19] reported that the leisure activity sector at Green Island in Taiwan consumes 21% of all energy used and generates 21% of all CO_2_ emissions there.

Researchers proposed several useful assessment tools by which to quantify tourism-related environmental loads. Prior studies on tourism-related CO_2_ emissions used a bottom-up approach involving industry and tourist analyses, such as life cycle assessment (LCA) [12,19], along with a top-down approach using environmental accounting, as with an input/output (I/O) model [23]. Emissions from tourism consumption-based accounting frameworks are calculated using the life cycle assessment (LCA) approach. Kuo et al. [19] applied the LCA approach to an inventory of island tourist data and demonstrated the feasibility of this methodology. Hence, industrial ecology may become an important tool in tourism management. An inventory of an entire tourist trip may be conducted using the LCA approach. In this manner, complete descriptions of the impact of tourism may be provided, and CO_2_ emissions derived from the different sectors can be compared [24]. One limitation of the LCA approach is that it is impractical to calculate data for all indicators in all sectors of a tourist trip. In recent years, a combination of the I/O model and the tourism satellite account (TSA) was applied, from the global to the city scale [25]. The top-down method requires that energy consumption and CO_2_ emission data are monitored at the national or regional level. Thus far, no statistical or monitoring system for GHG emissions has been designed for tourism development in the protected areas of China. For this reason, the present study adopted mainly the bottom-up approach to estimate the energy use and CO_2_ emissions associated with tourism.

### 2.2. Low-Carbon Behavior and Its Influence Factors

As global environmental challenges continue to increase, several researchers have begun to pay attention to low-carbon behavior and its influencing factors. Low-carbon behavior is a subset of pro-environmental behavior; the concept of low-carbon behavior is defined as the behavior of ensuring the sustainability of the ecosystem while maintaining normal economic development [26]. Stern [27] divided pro-environmental behavior into private-sphere behaviors, such as habitual and consumer behavior and recycling resources, and public-sphere behaviors, such as supporting environmental policies and encouraging others to participate in environmental activity. In tourism research, despite studies indicating that the degree of low-carbon behavior declines when people are on holiday [28], there is some evidence in experimental studies suggesting that certain tourists are concerned about global climate change and are willing to consider the environmental impact of their travel decisions [15].

To promote low-carbon behavior, several studies have endeavored to identify the factors effectively influencing environmental behavior, such as a low-carbon attitude, low-carbon knowledge, situational factors, and demographics [16,26,29]. The effects of attitude on low-carbon behavior are controversial. Low-carbon attitude might significantly promote low-carbon behavior. However, others believe that attitude has no significant bearing on behavior. Thus, further research is obviously needed. Chen and Li [15] reported that individuals with good environmental attitudes are more likely to participate in energy-conserving practices. Eijgelaar et al. [16] suggested that an environmentally friendly attitude does not necessarily result in the purchase of eco-labeled products. One study claimed that even the best consumer intentions may not suffice to convert a favorable attitude into pro-environmental behavior [28]. The relationship between low-carbon knowledge and behavior has been investigated using various research modes and dimensions. Expansion of the low-carbon knowledge of an individual is conducive to low-carbon behavior. Tourism research has shown that environmental knowledge is an antecedent to environmental attitude and may also affect future low-carbon behavior [6]. The relationships between situational factors, such as supportive policies, economic costs, social systems, social culture, and low-carbon behavior have been intensely debated. Stern [27] emphasized the importance of taxes, economic incentives, and other supportive policies in influencing low-carbon behavior. It remains to be established whether attitude, knowledge or situational factors strongly influence the low-carbon behavior of tourists, especially in the protected areas of China. Therefore, we examined the above-mentioned factors as antecedents to low-carbon behavior, thereby filling this research gap.

## 3. Methods

### 3.1. Study Area

The study focused on protected areas in the Qinling Mountains of China. The Qinling Mountains have high biodiversity and are a tourism hotspot in northern China. By 2011, eight national nature reserves had been established in the region and formed a protected area network. The national parks and tourism districts in the protected areas of the Qinling Mountains have impressive and varied scenery, attractive natural resources, and pristine wilderness areas. Tourism development has become a major policy of the local government, the objective being to increase community employment and boost local economic growth.

Tourism and the rich biodiversity and habitat vulnerability of the protected areas of the Qinling Mountains make this region ideal for the present research. The sample used in the analyses comprises three nature reserves and four national forest parks in and around the protected areas (Figure 2). The number of tourists in the study area increased from 836,400 in 2014 to ~1,968,000 in 2019. CO_2_ emissions from tourism may, therefore, become a serious environmental issue and should be proactively resolved.

### 3.2. Data and Variables

#### 3.2.1. Data Source

The survey data used here were derived from the Survey of Protected Areas and Tourism, conducted in 2014, 2016, and 2019. Following the peak tourism seasons in the study area, annual data were collected in early spring (February–April) and late summer (July–September). The field team gathered data by means of consulting tourists who had concluded their visits and were waiting in the rest area of the scenic region. Interviews were conducted in person to obtain accurate information. The duration of each interview was 40–70 min. A total of 1002 questionnaires were completed, of which 941 were deemed valid after filtration. Basic information on the tourists in the Qinling Mountains is summarized in Table 1 and Table 2. 

The questionnaire was tested before the actual data collection. The final version of the questionnaire was divided into respondent background information, type of transportation taken from the tourists’ places of residence to the destination, type of transportation used when at the destination, length of visit, recreational activity type, and low-carbon attitude and behavior of the tourists.

#### 3.2.2. Emission Factors

Based on regional characteristics and data availability, the emission factors given in the literature were used. The per capita data used in this study are shown in Table 3.

#### 3.2.3. Variable Selection


(1)Dependent variables


Behavior is the general term for a series of simple actions expressed in daily life [30]. Based on the existing literature [27], tourist low-carbon behavior refers to behaviors that positively influence substance and fossil energy consumption and alter the structure and dynamics of an ecosystem or the biosphere. Previous studies qualitatively measured low-carbon behavior. Barr et al. [29] suggested that habitual and consumer behaviors and recycling resources may be used to evaluate low-carbon behavior. Chen and Li [15] classified low-carbon behavior into private-sphere and public-sphere behaviors. Based on prior research and tourist characteristics in China, the present study evaluated low-carbon behavior using the CO_2_ emissions produced per tourist per trip to quantify the degree of pro-environmental behavior.
(2)Independent variables

According to the “theory of planned behavior” [31], the behavior of an individual is determined by the intention of the person to perform or not to perform an action. In turn, this intention is influenced by attitudes and subjective norms. The theory of reasoned action was extended to form the theory of planned behavior by introducing perceived behavioral control to measure the perception of the ability of an individual to perform a particular behavior. Low-carbon knowledge is based on the awareness of an individual of environmental issues and the behaviors that lead to them [15]. To explain complex behavior, situational factors were incorporated into the conventional “theory of planned behavior” model. Previous research showed that low-carbon behavior is a multidimensional structure encompassing low-carbon awareness and knowledge, personal and social norms, and situational factors [26]. Based on the findings from the literature review and the tourist characteristics in the study area, the independent variables were grouped into the low-carbon attitude, low-carbon knowledge, and situational factor categories.

*Low-carbon attitude.* Previous studies emphasized the impact of attitude on low-carbon behavior. It was widely believed that individuals with good low-carbon attitudes are relatively more likely to participate in energy conservation practices [32]. However, Eijgelaar et al. [16] argued that an environmentally friendly attitude does not necessarily result in a change in actual travel behavior. Rodríguez-Barreiro et al. [33] defined low-carbon attitudes according to the attitude formation, action intention, nature conservation view, and extension activity categories. In the present study, low-carbon intention, low-carbon preparation, and low-carbon loyalty were used to characterize each respondent’s low-carbon attitude. The low-carbon behavior intention of tourists may directly affect their low-carbon consumption behavior [26]. Initial travel decisions, such as seeking information about the potential environmental impact of a trip prior to booking, might play a critical role in the adoption of low-carbon behavior by tourists [28]. The level of low-carbon behavior generally rises when tourists are willing to pay the extra money and tolerate certain inconveniences for the sake of environmental protection while they are traveling [34]. Hence, the self-reported purchase of low-carbon products at comparatively higher prices was used to measure low-carbon loyalty in this instance.

*Low-carbon knowledge.* The impact of low-carbon knowledge on low-carbon behavior was highlighted in several studies. It was widely believed that increasing the awareness of low-carbon tourism will help tourists behave in an environmentally responsible manner once at their destinations [28]. Individuals can only consciously adopt pro-environmental behavior when they have a basic understanding of the low-carbon issues and behaviors that lead to environmental problems [15]. However, Kempton [35] suggested that according to both environmentalist and anti-environmentalist groups, environmental knowledge per se is not a prerequisite for pro-environmental behavior.

*Situational factors.* Situational factors are external and affect the low-carbon behavior of tourists. They include economic costs, social systems and culture, and administrative regulations [15]. Based on previous research and the tourist destination characteristics, environmental education, policy rewards, carbon labeling, and geographical environment were used to characterize the situational factors. Educating and informing tourists regarding the negative environmental impact of travel has been proposed as a way to modify their behavior [28]. An empirical study explored the impact of visual attention to carbon labels on purchasing behavior in tourism [36]. Stern [27] emphasized the importance of supportive policies, such as subsidies, in influencing low-carbon behavior.
(3)Control variables

Studies have shown that education and income level, age, and gender significantly influence low-carbon behavior [15,26,33]. Thus, the present study integrated the control variables affecting low-carbon behavior into the demographic characteristics of tourists. Definitions of the variables in this study are shown in Table 4.

### 3.3. Research Methods

#### 3.3.1. Evaluation of Energy Use and CO_2_ Emissions from Tourism

Comprehensive calculations of tourism-derived GHG emissions are complex, especially when the emission data must be compared against economic data to obtain GHG intensity data [37]. Tourism is a major contributor to climate change as it consumes fossil fuels and emits CO_2_, a greenhouse gas [13]. Based on previous studies, tourist consumption is categorized by transportation, accommodation, recreational activity, and food [38]. To simplify the analysis here, food production and consumption were excluded as they are difficult to measure with a survey. Therefore, the present study examined emissions related to travel to, at, and from a destination, as well as accommodation and recreational activity when at the destination.

Life cycle assessment (LCA) is a bottom-up analysis that can assess the entire carbon footprint, from resource extraction to end use [37]. Hence, the present study used LCA to quantify energy use and CO_2_ emissions from tourists traveling to, at, and from their destination. Compared with the ecological footprint and eco-efficiency analyses, LCA is a comprehensive assessment tool that enables an inventory of the full environmental impact of tourism. According to the relevant literature [10], the LCA-related formulae are expressed as follows:(1)$$\mathrm{LE}=\overset{n}{\underset{i=1}{\sum }}P\times {\mathrm{le}}_{i}$$
(2)$$\mathrm{TE}=\overset{n}{\underset{i-1}{\sum }}P_{i}D_{i}\beta_{i}$$
where LE is the total energy-related CO_2_ emission from each tourism category, P is the scale of a specific tourism activity (hotel, transport, traveling, and recreation), le*_i_* is the fossil energy consumption/CO_2_ emission per unit of a specific tourism activity, TE is the total fossil energy consumption/CO_2_ emission by different vehicles, P*_i_* indicates the total number of tourists in transport mode *i*, D*_i_* is the distance of transport mode *i* (km), and β*_i_* is the fossil energy consumption/CO_2_ emission factor for transport mode *i*.

#### 3.3.2. Estimation Method

The degree of tourist low-carbon behavior was divided into five different categories. An ordinal logistic regression model was used to determine how the variables jointly affect the low-carbon behavior of tourists in the protected areas of China. This technique handles the outcome variables with >2 ordered categories. Here, the dependent variable was CO_2_ emissions per tourist per trip. The independent variables were low-carbon awareness, low-carbon knowledge, situational factors, and demographic characteristics. The estimated ordinary logistic model is expressed as follows:(3)$$Logit\left(P_{ih}\right)=\ln\frac{P_{ih}}{1-P_{ih}}=a_{h}+b_{1h}X_{i1}+b_{2h}X_{i2}+\cdots +b_{kh}X_{ik}=\overset{k}{\underset{j=0}{\sum }}b_{jh}X_{ij}$$
where *P_ih_* = *P* (*y_i_* ≤ *h*), *i* and *h* are the *i*th estimates from the *h*th study, *a_h_* is a constant, and *b*_1*h*_, *b*_2*h*_, …, *b*_*kn*_ are the estimated coefficients for the explanatory variables.

The unobserved heterogeneity omitted the variable bias, and measurement error may occur in both the independent and control variables and may make it difficult to obtain consistent estimates. Therefore, a series of robustness tests were conducted to validate the credibility of the empirical results. OLS regression and Oprobit were used for the comparative analyses, and they partially demonstrated the robustness of the results, to a certain extent.

## 4. Results

### 4.1. Energy Use and CO_2_ Emissions among the Different Tourism Sectors

According to the LCA, the emissions from tourism were estimated to rise by 3.6% per year and reach 1.74 × 10^11^ g CO_2_ by 2019 (Figure 3)—almost doubling in the previous 5 years. Based on the study area survey data from the Nature Reserve Administration, the number of tourists in the study area rose by 56.53% between 2014 and 2019. This increase was a key contributor to rising tourism-related CO_2_ emissions. CO_2_ emissions per tourist per trip increased from 54.13 kg in 2014 to 80.68 kg in 2019 and the energy use was in the range of 960.28–1363.27 MJ. The energy use and CO_2_ emissions for different tourism sectors in the study areas in 2014, 2016, and 2019 are shown in Table 5 and Figure 4.

Transportation between the points of origin and the destinations accounted for the largest proportion of the total fossil energy consumption (1.44 × 10^9^ in 2014, 1.72 × 10^9^ in 2016, and 2.06 × 10^9^ in 2019) and generated the highest CO_2_ emission levels (7.66 × 10^10^ in 2014, 8.46 × 10^10^ in 2016, and 13.9 × 10^10^ in 2019). Statistical analysis of the basic characteristics of the questionnaire revealed that most tourists traveled by private vehicles to the protected areas of the Qinling Mountains. This mode of transportation accounted for 39.48% of the annual energy use (8.14 × 10^8^ MJ) and 38.66% of the CO_2_ emissions (5.39 × 10^10^ g) in 2019, followed by trains (3.63 × 10^10^ g; 26.01%) and buses (3.57 × 10^10^ g; 25.62%).

Private vehicles consumed the most energy (48.9%), followed by buses (23.28%). Private vehicles also contributed a major part of the total CO_2_ emissions (44.57% in 2014, 43.85% in 2016, and 49.15% in 2019). The total CO_2_ emissions from the vehicles used at the tourist destinations increased from 5.34 × 10^9^ g in 2014 to 15.1 × 10^9^ g in 2016 but decreased to 9.63 × 10^9^ g in 2019.

Most tourists (75.81%) stayed in hotels in the Qinling Mountains. Nevertheless, the energy intensities of the star-rated and budget hotels were higher than those of other types of accommodations. The accommodation sector produced 2.04 × 10^10^ g CO_2_ emissions in 2019. The hotel sector (star-rated and budget hotels) consumed the most energy (92.12%) and generated 84.18% of the total CO_2_ emissions. Similar results were obtained in 2016. In that year, most accommodation fossil energy consumption was linked to “budget hotels” (63.48%) and “star-rated hotels” (19.82%).

The proportions of energy use and CO_2_ emissions did not significantly differ among recreational activities in all the surveyed years. The energy intensity of the motorized water activities was the highest (236.8 MJ/visitor). They consumed 4.05 × 10^7^ MJ (40.56%) of the total energy and generated 26.20 × 10^8^ g of CO_2_ emissions in 2019. Although 88.54% of the tourists went sightseeing, the energy use and CO_2_ emission levels of this particular recreational activity were relatively low (4.24 × 10^6^ MJ, 4.19%; and 2.10 × 10^8^ g, 4.61%). The energy intensity of sightseeing is also comparatively low (8.5 MJ/visitor).

### 4.2. Regression Results

Table 6 lists the results of the regressions on the factors influencing low-carbon behavior. The OLS (ordinary least squares) and Oprobit models were used here to re-estimate the results, determine whether the estimates were affected, validate the robustness of the Ologit regression estimation results, and demonstrate that they are stable. The re-estimated results were nearly consistent with the Ologit regression, in terms of the signs and significance levels of the variables. Hence, the estimated results were robust in terms of the estimation method.

Low-carbon attitudes significantly motivate tourists to behave in a low-carbon manner. The results of this study indicated that low-carbon behavioral intentions on holiday are positively correlated with individual CO_2_ emissions and the findings were consistent with the empirical literature [26]. The present study found that self-reported quests for information about the environmental impact of a trip prior to booking are significantly and positively associated with low-carbon behavior and this was consistent with the findings of Juvan and Dolnicar [28]. Initial travel decisions usually determine the mode of transport required to reach the tourist destination and also the choice of accommodation. Both of the foregoing factors contribute the most to tourism-related CO_2_ emissions. Low-carbon loyalty is characterized by the willingness to pay extra and/or tolerate certain inconveniences for the benefit of environmental protection. It helps convert the intention of tourists into action and was reported in the literature by Cherrier et al. [34]. The results of the present study corroborated the aforementioned concepts and indicated that low-carbon products will be purchased if required, while higher prices will significantly influence the low-carbon behavior of tourists during their travels.

The model herein had four low-carbon knowledge variables. Tourists who are familiar with low-carbon behaviors and know how to conduct them on their trip can increase their level of actual low-carbon behavior. This finding was consistent with those of McDonald et al. [39], who concluded that low-carbon knowledge is the key to changing tourists’ attitudes from those who intend to behave in a low-carbon manner to those that actually do behave that way. The results here indicated that tourists can only consciously adopt low-carbon behaviors once they have a basic understanding of the environmental issues and behaviors that lead to fossil energy consumption and CO_2_ emissions during their travels.

The estimates in the present study indicated that environmental education has a significant positive impact in terms of regulating low-carbon behavior. Exposure to environmental messages from traditional media and the internet can positively influence the adoption of environmental actions such as green consumption and resource conservation [40]. Hence, tourists who have been educated and informed about the negative environmental impact of travel will engage in relatively higher levels of low-carbon behavior. The present study provided strong evidence that policy reward is significantly and positively associated with increased low-carbon behavior. Thus, policy rewards motivate tourists to adopt environmentally friendly practices when they make travel decisions. This finding supports the empirical results of Chen and Li [15] and Stern [27].

The present study also showed that women had comparatively superior low-carbon behavior performance. This observation was consistent with the report of Yang et al. [26]. Low-carbon behavior performance tended to improve with income level. This finding aligned with the conclusions of Chen and Li [15].

## 5. Discussion and Conclusions

With China already committing to lowering peak carbon dioxide emissions before 2030 and achieving carbon neutrality before 2060, low-carbon transformation in all industries should be required in pursuit of this goal. Chinese low-carbon transformation, or sustainable development, requires more in terms of government regulation [41,42]. The tourism sector is vulnerable to the effects of climate change and yet, paradoxically, contributes to the emission of GHGs that cause global warming. The transformation of tourism to a low-carbon industry is not only necessary for its own sustainable development but also part of the more general mission of responding to global climate change and achieving carbon peaking and carbon neutrality goals. To develop low-carbon tourism in protected areas, planners must understand the energy requirements and carbon emissions of nature-based tourism, as well as the factors that motivate tourists to adopt low-carbon behavior. The present study used life cycle assessment (LCA) to investigate the energy use and CO_2_ emission associated with tourism and identify the factors influencing low-carbon behavior in tourists. To this end, survey data from 2014 to 2019 for the protected areas of the Qinling Mountains of China were used. The results showed that energy use and CO_2_ emission have increased over time in various tourism sectors. Moreover, energy use and CO_2_ emission per tourist, per trip, also increased in the study area between 2014 and 2019. Tourism-related transportation accounted for the majority of annual energy use and CO_2_ emissions (72.6% and 79.7%, respectively). This finding is consistent with those of Kuo and Chen [24], who found that the transportation sector consumes the largest amount of energy (67%) and generates the largest proportion of CO_2_ emissions (68%) among the different tourism sectors in Taiwan’s Penghu Islands region in China. According to Kuo et al. [24], the per-day energy usage of an average domestic tourist in the Penhu Island is 1606 MJ, while in the Kinmen Island, it is 1387 MJ. This implies that tourists in Qingling Mountains in 2014, 2016, and 2019 used less energy than the Penhu Island tourists and Kinmen Island tourists. However, protected areas have a policy priority for environmental protection, and thus they may be more likely to embrace sustainable development ideas than other “non-protected” areas [43]. Hence, local governments should propose effective strategies to encourage the inclusion of sustainable tourism ideas in protected area policies. The present study used the calculations of CO_2_ emissions per tourist, per trip, to examine the factors that might significantly influence the low-carbon behavior described in the literature. The regression results showed that low-carbon attitude, low-carbon knowledge, environmental education, policy rewards, and income all strongly influence the low-carbon decisions made by tourists.

In a new development paradigm, with domestic circulation as the mainstay and the domestic and international calculations reinforcing each other, Chinese tourism in the post-epidemic era is bound to offer a breakthrough in terms of unleashing the potential of domestic demand [44]. Therefore, this public-health emergency can provide a buffer period for the tourism industry to adjust its product structure and improve development quality according to the concept of low-carbon tourism development. Most importantly, the reduction of the carbon footprint associated with tourism helps maintain a healthy ecological environment at tourist destinations, through the practice of low-carbon behaviors, and directly affects the sustainable development of tourism. Therefore, the findings of the present study have important policy implications for the tourism industry, as well as for the more expansive goals of curtailing carbon emissions and establishing and maintaining carbon neutrality in China.

Government departments should improve the policy system and standard system of low-carbon tourism, as well as the digital system for controlling carbon emissions, such as mastering the carbon emission of tourism via a census, formulating industry standards for the carbon emission reduction of tourism, promoting tourism enterprises to actively participate in carbon market trading, and realizing the carbon neutralization of tourism through an offset mechanism. As the world’s largest domestic tourism market, the state of China’s tourism industry is particularly urgent in terms of carbon emission reduction. Assessing tourism fossil energy consumption and carbon emissions is the basis for developing green and low-carbon tourism. The present study shows that the protected areas attract a large number of tourists drawn in by short tours and flash tours, which, in turn, contribute to fossil energy consumption and carbon emissions. In the future development of low-carbon tourism, the government and enterprises should fully accept that the connection between the tourist source and the destination is no longer simply a matter of capital transfer between economically developed and less developed regions, and the relationship between the carbon sink and carbon source should also be considered from the ecological perspective. Therefore, government departments should speed up the formulation of industry standards for carbon emission reductions in tourism, such as adding the relevant carbon emission indicators to the assessment standards of starred hotels. The government should provide tax breaks and concessions for low-carbon tourism enterprises and levy additional taxes and fees on “three-high” tourism enterprises (those responsible for high pollution, high fossil energy consumption, and high emissions). In terms of financing, preferential policies should be adopted to encourage the low-carbon development of tourism enterprises and build low-carbon tourist attractions. In July 2021, China’s unified national carbon emissions trading market went online. The government should introduce relevant policies to encourage tourism enterprises to take advantage of this platform to actively participate in carbon market trading, such as by choosing voluntary carbon-trading projects to achieve carbon neutrality through the offset mechanism.

Policy interventions should target energy conservation and emissions reduction in the development of tourism in protected natural areas. The findings of the present study indicate that the protected areas of the Qinling Mountains of China are being challenged by climatic unsustainability. Tourism substantially increased the carbon emissions in the study region between 2014 and 2019. One recommendation for the sustainable development of tourism is to convert from traditional high fossil energy consumption to efficient and clean energy in the entertainment facilities found in the protected regions. Popular powered leisure and recreation activities might emit more carbon than traditional sightseeing as the former use energy-intensive equipment. The present study demonstrated that the focus on tourism activity in protected areas invariably leads to increased energy demand. Tourist destinations that specialize in sightseeing and attractions are less likely to increase their annual fossil energy consumption. However, the focus of the tourism market has turned toward other activities that require considerable amounts of energy in protected areas. Hence, energy-intensive and adventure activities should be minimized in and around protected areas to help establish and maintain environmentally sustainable tourism. All tourist resorts, A-level scenic spots, and star hotels should create more high-quality ecotourism routes and “near-zero carbon emission” demonstration projects, to ensure that low-carbon products are fully available in tourism destinations and all aspects of tourism, such as diet, accommodation, and garbage disposal.

Mitigation policies for sustainable tourism development should concentrate on changing the behavior of tourists so that they actively reduce their own carbon emissions. Tourists are the main energy users and carbon emitters in protected areas and should be encouraged to switch to low-carbon consumption patterns in terms of transportation and accommodation. The latter are the major sources of tourist energy use and CO_2_ emissions. Thus, mitigation policies in the tourism sector should address changes in consumption and guide tourists toward the use of low-carbon products. Travel frequency and distance, as well as the mode of transportation, markedly influence tourism-related carbon emissions. The rates of carbon consumption and emissions are increasing faster than tourism development. Therefore, energy-efficient transportation to and from tourist destinations should be advocated. These methods of transportation could include electric trains and buses for local travel. The use of private vehicles is becoming ever more popular for visiting nature-based attractions in protected areas. For this reason, tourists should be encouraged to visit nearby iconic attractions or travel long distances using only energy-sparing vehicles. Low-carbon accommodation facilities should be promoted by reducing living spaces and using disposable hotel supplies.

Fiscal or financial policies, such as taxation and subsidies, should be used to guide the low-carbon behavior of tourists. However, there are still relatively few tourists who are intrinsically motivated to behave in an environmentally friendly manner. It is still unlikely that many tourists will change their behavior to help reduce carbon emissions when they plan holiday travel [28]. The present study revealed that policy reward has a positive impact on the low-carbon behavior of tourists. Consequently, government agencies and social organizations should use fiscal and financial policies to incentivize travelers to make ecologically responsible choices, such as choosing to use public transportation, joining environmentally friendly tours and tourism activities, and so on.

Policymakers should exploit both the traditional media and the internet to raise awareness in tourists concerning low-carbon behaviors in their travel decision-making. Theoretically, it is not enough to transform low-carbon behavior into self-aware habits in tourism merely by force and by offering economic incentives; there should also be the guidance of social ethics. Environmental education can play a role in promoting the formation of tourists’ habits. The present study showed that the implementation of low-carbon behavior increases with increased low-carbon knowledge and environmental education levels, and this may help improve sustainable tourism in protected areas. Therefore, local governments with jurisdiction over protected areas should increase their environmental publicity and enhance tourist awareness of the impact of low-carbon behavior on environmental quality. Moreover, the government should take the initiative to fully infiltrate environmental education into the national education system, in order to stimulate the public’s willingness and spontaneous action regarding low-carbon tourism. The Internet and intelligent technology should be used to reduce the action cost of low-carbon tourism and improve the benefits of tourists’ green behaviors, so as to create synergy with environmental education and improve the effects of policy implementation.

## Figures and Tables

**Figure 1 ijerph-20-01277-f001:**
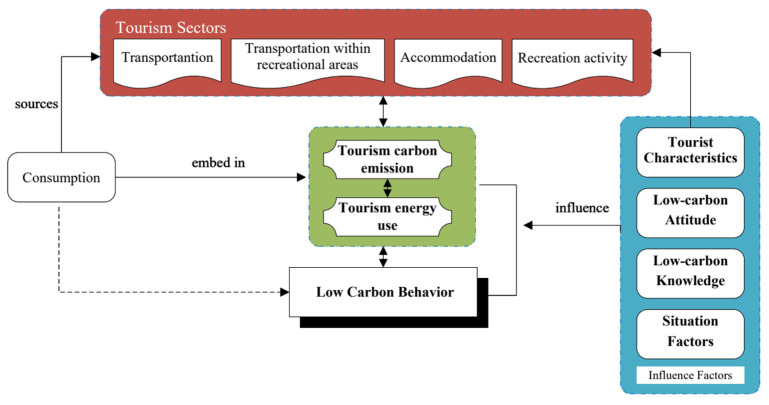
Theoretic analysis framework.

**Figure 2 ijerph-20-01277-f002:**
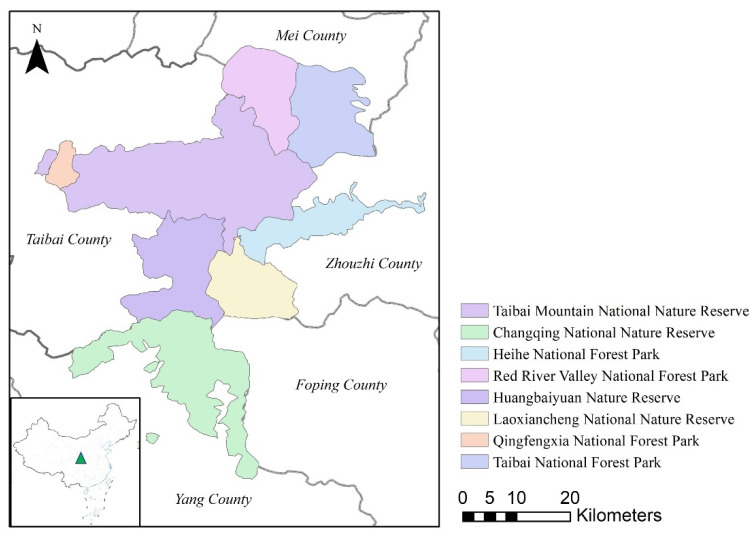
Study area.

**Figure 3 ijerph-20-01277-f003:**
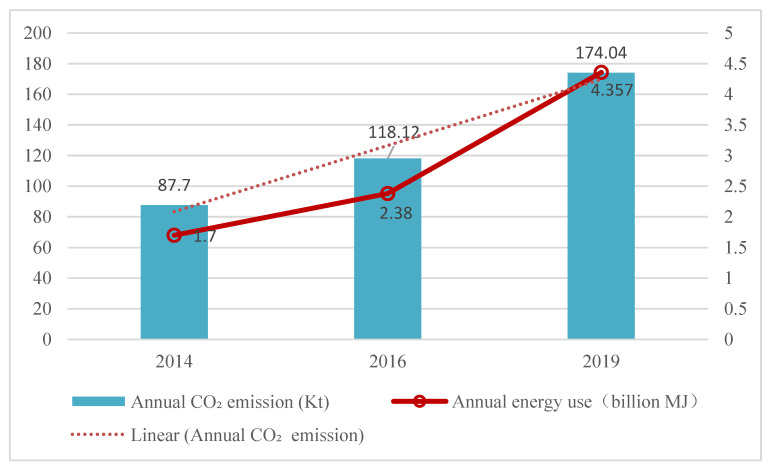
Annual energy uses and CO_2_ emissions of tourism in the study area.

**Figure 4 ijerph-20-01277-f004:**
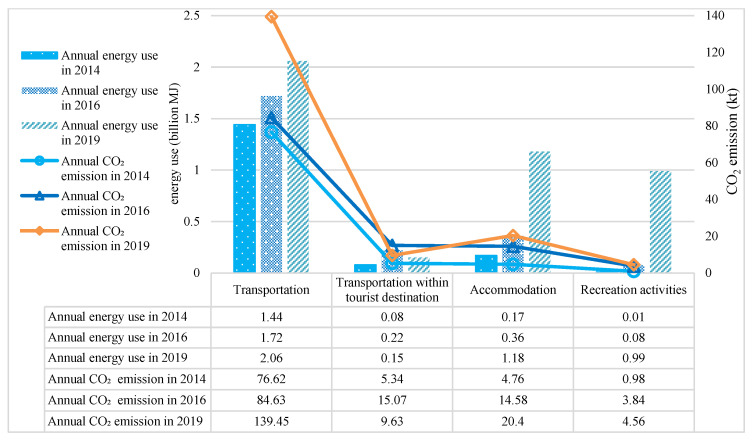
Annual energy uses and CO_2_ emissions of different sectors (2014, 2016, and 2019).

**Table 1 ijerph-20-01277-t001:** The basic information given by tourists to the Qinling Mountains.

Category	2014 (%)	2016 (%)	2019 (%)
1. Places of residence			
From Shaanxi province	78.6	70.65	69.12
Not from Shaanxi province	21.4	29.35	30.88
2. Length of stay (days)			
1	26.56	30.72	48.44
2	27.22	32.58	22.04
3	20.01	17.97	5.82
4	16.91	7.61	6.65
≥5	9.3	11.11	17.05
3. Type of transportation (from the tourists’ places of residence to the tourist destination)			
Airplane	3.33	3.19	3.16
Train	11.57	6.71	12.85
Private car	50.87	63.26	59.29
Bus	32.65	24.28	23.91
Motorcycle	1.59	2.56	0.79
4. Type of transportation within the recreational areas			
Motorcycle	8.49	2.77	3.35
Private car	39.46	44.61	46.64
Rental car	13.04	13.33	7.91
Bus	25.74	27.77	19.57
Small shuttle bus	32.67	33.69	22.53
Bicycle	9.63	8.12	0
5. Type of recreation activity			
Sight-seeing/landscape visiting	41.24	48.54	52.37
Historic site visiting/battle-site visiting)	16.42	13.11	18.97
Adventure recreation (e.g., rafting, rock climbing)	13.50	14.08	6.52
Motorized water activity	3.28	5.34	12.65
Nature recreation (e.g., swimming, fishing, viewing wildlife in its natural setting)	17.15	16.99	10.28

**Table 2 ijerph-20-01277-t002:** Percentage (%) of stays at different types of accommodation in the survey areas.

Number of Nights	Star-Rated Hotel (%)			Budget Hotel (%)			Rural Home Inns (%)			Private Home (%)		
	2014	2016	2019	2014	2016	2019	2014	2016	2019	2014	2016	2019
1	36.81	25.00	16.67	30.13	31.09	32.45	44.76	36.04	26.87	45.36	42.86	46.34
2	33.14	42.86	33.33	41.28	40.34	30.46	27.65	36.94	41.04	29.52	38.10	24.39
3	25.43	25.00	21.67	19.76	15.13	16.56	12.81	21.62	17.16	18.21	14.29	17.07
≥4	4.62	7.14	28.33	8.83	13.45	20.53	14.87	5.41	14.93	6.91	4.76	12.20

**Table 3 ijerph-20-01277-t003:** The per capita data used in this research.

Category of Loads	Energy Intensity Demand Factor	CO_2_ Emission Factor	Data Source
1. Type of transportation (tourist-generating region—tourist destination)			
Airplane	2.576 MJ/rpk ^a^	69 g/MJ ^b^	2001 [22], 2012 [19], 2011 [10]
Train	1.44 MJ/pkm ^c^	62 g/pkm	
Private car	1.03 MJ/pkm	68.7 g/pkm	
Bus	1.01 MJ/pkm	69.2 g/pkm	
2. Types of transportation within recreational areas			
Motorcycle	1.22 MJ/vkm	58 g/pkm	2001 [22],2015 [11], 2012 [19],2011 [10]
Private car	1.03 MJ/pkm	68.7 g/pkm	
Rental car	1.06 MJ/pkm	63 g/pkm	
Bus	1.01 MJ/pkm	69.2 g/pkm	
Small shuttle bus	0.59 MJ/pkm	51 g/pkm	
3. Energy intensity and CO_2_ emission of different accommodation types			
Star-rated hotel	155 MJ/visitor night	7900 g/visitor night	2001 [22], 2012 [19]
Budget hotel	110 MJ/visitor night	4140 g/visitor night	
Rural home inns	41 MJ/visitor night	1619 g/visitor night	
Private home	41 MJ/visitor night	1619 g/visitor night	
4. Energy intensity and CO_2_ emission of different recreation activities			
Sight-seeing	8.5 MJ/visitor	417 g/visitor	2001 [22], 2006 [12], 2012 [19]
Historic sites visiting	3.5 MJ/visitor	172 g/visitor	
Landscape watching	8.5 MJ/visitor	417 g/visitor	
Adventure recreation	35.1 MJ/visitor	2240 g/visitor	
Motorized water activity	236.8 MJ/visitor	15,300 g/visitor	
Nature recreation	70 MJ/visitor	1674 g/visitor	

^a^ rpk: revenue passenger-kilometers; ^b^ MJ: megajoule; ^c^ pkm: per passenger–kilometers.

**Table 4 ijerph-20-01277-t004:** Definitions of the variables.

Variables	Variable Description	Mean	SD
Behavior	The degree of low-carbon behavior of tourists on this trip	3.00	1.41
LAT-intention	Intentions of low-carbon behavior = 1 and 0 otherwise	0.69	0.46
LAT-preparation	Looking for information about the environmental impact of this trip before booking = 1 and 0 otherwise	0.35	0.48
LAT-loyalty	Low-carbon products will be purchased if required at a higher price = 1 and 0 otherwise	0.49	0.50
KNO-behavior	Know which behaviors are low-carbon behaviors on a trip	0.64	0.48
KNO-conduct	Know how to perform low-carbon behavior on a trip	0.56	0.49
KNO-importance	Know the importance of low-carbon behavior	0.67	0.47
KNO-footprint	Tourists have ever heard of a vacation’s carbon footprint or carbon calculators = 1 and 0 otherwise	0.48	0.49
Environmental education	Tourists have been educated and informed about the negative environmental impact of travel = 1 and 0 otherwise	0.67	0.47
Carbon label	Self-reported paying attention to the energy labels or carbon labels	0.09	0.27
Policy rewards	If there are policy rewards, low-carbon behavior will be carried out in this trip = 1 and 0 otherwise	0.63	0.48
Geographical environment	Tourist destinations located in protected areas have an impact on low-carbon behavior = 1 and 0 otherwise	0.55	0.57
Education	Level of education	2.09	1.05
Income	Household annual income	2.24	1.14
Age	The age of the respondent	1.83	0.86
Gender	Female = 0, Male = 1	0.47	0.49

**Table 5 ijerph-20-01277-t005:** Annual energy uses and CO_2_ emissions of different sectors.

Tourism Sectors		Annual Energy Use (MJ)			Annual CO_2_ Emission (g)			Energy Use Per Tourist Per Trip (MJ)			CO_2_ Emissions Per Tourist Per Trip (g)		
		2014	2016	2019	2014	2016	2019	2014	2016	2019	2014	2016	2019
Transportation	Airplane	2.57 × 10^8^ (17.86%)	4.37 × 10^8^ (25.36%)	3.77 × 10^8^ (18.31%)	6.89 × 10^9^ (9.00%)	1.17 × 10^10^ (13.83%)	1.35 × 10^10^ (9.70%)	644.64	747.11	902.49	38,481.74	41,237.71	56,487.87
	Train	4.35 × 10^8^ (30.14%)	4.71 × 10^8^ (27.30%)	3.38 × 10^8^ (16.43%)	1.87 × 10^10^ (24.42%)	2.03 × 10^10^ (23.94%)	3.63 × 10^10^ (26.01%)						
	Private car	1.59 × 10^8^ (11.02%)	7.02 × 10^8^ (40.72%)	8.14 × 10^8^ (39.48%)	1.06 × 10^10^ (13.83%)	4.51 × 10^10^ (53.28%)	5.39 × 10^10^ (38.66%)						
	Bus	5.88 × 10^8^ (40.78%)	1.0 × 10^8^ (6.12%)	5.31 × 10^8^ (25.75%)	4.03 × 10^10^ (52.57%)	0.72 × 10^10^ (8.48%)	3.57 × 10^10^ (25.62%)						
	Motorcycle	2.83 × 10^6^ (0.20%)	8.80 × 10^6^ (0.51%)	0.57 × 10^6^ (0.03%)	1.35 × 10^8^ (0.18%%)	3.99 × 10^8^ (0.47%)	0.27 × 10^8^ (0.02%)						
	Subtotal	1.44 × 10^9^	1.72 × 10^9^	2.06 × 10^9^	7.66 × 10^10^	8.46 × 10^10^	13.9 × 10^10^						
Transportation within the tourist destination	Motorcycle	7.80 × 10^6^ (9.76%)	4.89 × 10^6^ (2.20%)	4.33 × 10^6^ (2.99%)	3.70 × 10^8^ (6.93%)	2.33 × 10^8^ (2.20%)	2.06 × 10^8^ (2.14%)	95.53	120.72	112.34	6384.51	8653.27	8536.58
	Private car	3.58 × 10^7^ (44.81%)	9.70 × 10^7^ (43.64%)	7.09 × 10^7^ (48.90%)	2.38 × 10^9^ (44.57%)	6.61 × 10^9^ (43.85%)	4.73 × 10^9^ (49.15%)						
	Rental car	1.05 × 10^7^ (13.14%)	4.44 × 10^7^ (19.99%)	2.71 × 10^7^ (18.67%)	6.24 × 10^8^ (11.69%)	2.64 × 10^9^ (17.52%)	1.61 × 10^9^ (16.72%)						
	Bus	1.53 × 10^7^ (19.15%)	5.26 × 10^7^ (23.65%)	3.37 × 10^7^ (23.28%)	1.05 × 10^9^ (19.66%)	3.60 × 10^9^ (23.90%)	2.31 × 10^9^ (24.03%)						
	Small shuttle bus	1.05 × 10^7^ (13.14%)	2.34 × 10^7^ (10.52%)	0.90 × 10^7^ (6.17%)	9.08 × 10^8^ (17.15%)	1.99 × 10^9^ (13.19%)	0.77 × 10^9^ (7.96%)						
	Subtotal	7.99 × 10^7^	2.22 × 10^8^	1.45 × 10^8^	5.34 × 10^9^	15.1 × 10^9^	9.63 × 10^9^						
Accommodation	Star-rated hotel	2.66 × 10^7^ (15.29%)	7.12 × 10^7^ (19.82%)	7.73 × 10^8^ (65.71%)	1.35 × 10^9^ (28.36%)	3.63 × 10^9^ (24.89%)	7.91 × 10^9^ (38.71%)	189.78	238.34	281.00	7918.65	9866.72	12,231.95
	Budget hotel	1.25 × 10^8^ (71.84)	2.28 × 10^8^ (63.48%)	3.11 × 10^8^ (26.41%)	2.48 × 10^9^ (52.17%)	8.58 × 10^9^ (58.86%)	9.30 × 10^9^ (45.47%)						
	Rural home inns	1.67 × 10^7^ (9.65%)	4.90 × 10^7^ (13.65%)	7.07 × 10^7^ (6.01%)	7.03 × 10^8^ (14.87%)	1.94 × 10^9^ (13.28%)	2.52 × 10^9^ (12.34%)						
	Private home	5.56 × 10^6^ (3.22%)	1.10 × 10^7^ (3.05%)	2.07 × 10^7^ (1.87%)	2.19 × 10^8^ (4.60%)	4.33 × 10^8^ (2.97%)	6.82 × 10^8^ (3.49%)						
	Subtotal	1.74 × 10^8^	3.59 × 10^8^	11.76 × 10^8^	4.76 × 10^9^	1.46 × 10^10^	2.04 × 10^10^						
Recreation activities	Sight-seeing	1.97 × 10^6^ (12.23%)	3.55 × 10^6^ (4.19%)	4.24 × 10^6^ (4.19%)	9.66 × 10^7^ (9.87%)	1.76 × 10^8^ (4.58%)	2.10 × 10^8^ (4.61%)	30.33	54.74	67.44	1349.61	2674.23	3419.49
	Historic sites visiting	7.04 × 10^5^ (4.37%)	7.51 × 10^5^ (0.89%)	9.21 × 10^5^ (0.92%)	3.46 × 10^7^ (3.53%)	3.54 × 10^7^ (0.92%)	4.35 × 10^7^ (0.95%)						
	Landscape viewing	6.84 × 10^5^ (4.24%)	1.38 × 10^6^ (1.63%)	1.67 × 10^6^ (1.67%)	3.35 × 10^7^ (3.42%)	6.96 × 10^7^ (1.81%)	8.38 × 10^7^ (1.84%)						
	Adventure recreation	5.44 × 10^6^ (33.74%)	1.14 × 10^7^ (13.59%)	1.37 × 10^7^ (13.73%)	3.47 × 10^8^ (35.42%)	7.34 × 10^8^ (19.10%)	8.76 × 10^8^ (19.22%)						
	Motorized water activity	3.7 × 10^6^ (23.36%)	3.44 × 10^7^ (40.64%)	4.05 × 10^7^ (40.56%)	2.43 × 10^8^ (24.83%)	22.20 × 10^8^ (57.86%)	26.20 × 10^8^ (57.48%)						
	Nature recreation	3.56 × 10^6^ (22.07%)	3.30 × 10^7^ (39.06%)	3.89 × 10^7^ (38.88%)	2.25 × 10^8^ (22.93%)	6.04 × 10^8^ (15.73%)	7.24 × 10^8^ (15.89%)						
	Subtotal	1.61 × 10^7^	8.46 × 10^7^	10.00 × 10^7^	9.80 × 10^8^	3.84 × 10^9^	4.56 × 10^9^						
Total		1.71 × 10^9^	2.39 × 10^9^	3.48 × 10^9^	8.77 × 10^10^	1.18 × 10^11^	1.74 × 10^11^	960.28	1160.91	1363.27	54,134.51	62,431.93	80,675.89

**Table 6 ijerph-20-01277-t006:** Factors that influence tourists’ low-carbon behaviors.

		Ologit	OLS	Oprobit
Attitude	LAT-intention	1.172 *** (0.148)	0.822 *** (0.101)	0.667 (0.087) ***
	LAT-preparation	0.316 ** (0.138)	0.237 ** (0.096)	0.185 (0.082) **
	LAT-loyalty	0.369 *** (0.129)	0.238 *** (0.090)	0.209 (0.076) ***
Knowledge	KNO-behavior	0.248 * (0.135)	0.192 ** (0.094)	0.156 (0.008) *
	KNO-conduct	0.414 *** (0.131)	0.296 *** (0.091)	0.229 (0.078) ***
	KNO-importance	0.080 (0.144)	0.050 (0.100)	0.021 (0.086)
	KNO-footprint	0.017 (0.131)	0.056 (0.091)	0.034 (0.078)
Situational	Environmental education	0.482 *** (0.141)	0.322 *** (0.098)	0.286 (0.084) ***
	Carbon label	−0.328 (0.215)	−0.225 (0.149)	−0.180 (0.128)
	Policy reward	0.626 *** (0.139)	0.417 *** (0.096)	0.353 (0.083) ***
	Geographical environment	−0.202 (0.126)	−0.125 (0.088)	−0.109 (0.075)
Controls	Education	0.104 (0.064)	0.075 * (0.045)	0.007 (0.038) *
	Income	0.178 *** (0.061)	0.124 *** (0.042)	0.100 (0.036) ***
	Age	0.038 (0.078)	0.042 (0.054)	0.027 (0.046)
	Gender	−0.234 * (0.131)	−0.165 * (0.091)	−0.153 (0.077) **
	_cons		1.067 *** (0.205)	
	F		15.897 ***	
	R^2^		0.230	
	Pseudo R^2^	0.079		0.075

Note: ***, **, and * indicate passing the significance test at 1%, 5%, and 10%, respectively.

## Data Availability

Data are contained within the article.

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
