# Peer review of "Measurement of Tourism-Related CO2 Emission and the Factors Influencing Low-Carbon Behavior of Tourists: Evidence from Protected Areas in China"

_ijerph, 2023, doi:10.3390/ijerph20021277_

Round 1
Reviewer 1 Report
This article is well written, has strong language expression ability, and has a lot of work. It is very meaningful to use questionnaire survey and data search to search data such as tourist behavior in China's Qinling Natural Reserve, which provides new ideas for carbon emission assessment and low-carbon emission reduction policy formulation of tourist destinations. This article is very suitable for publication in the IJERPH journal, but before publication, I suggest that the manuscript can be returned to the author for careful minor modifications, which is crucial for the successful publication of this article.
Some Comments
1. As for the 8 national nature reserves in Qinling Natural Reserve of China, the author should elaborate their geographical location and other information. Please try to draw a map to show them.
2.The format of references in line 317 is incorrect, please correct
3. Please specify the writing format of CO2. 2 You need to set the subscript form, such as line 381 and Figure.2.
4. Please correct the format of 383 rows and 384 rows. Table 5 should start a new row.
5. Please standardize the format of Table 5 and try to keep some data in one row.
6. More references need to be introduced in the discussion part. At present, there are few references in the whole discussion part, and some need to be cited to support key points. In addition, the author should try his best to compare the research results of Qinling Natural Reserve with those of other regions based on the research status at home and abroad.
7. Please standardize the reference format.
Author Response
Measurement of tourism-related CO2 emission and the factors influencing low-carbon behavior of tourists: Evidence from protected areas in China
We appreciate the comments and suggestions from the editor and reviewers. We have sincerely considered all the comments and revise the paper accordingly. For the detailed corrections, please see the revised manuscript, in which we highlight the major revision in Red. We also provide this point-by-point response to all the questions raised by the reviewers, in which our responses are highlighted in Blue while the reviewers’ and editor’s comments are kept in black.
Reviewer #1
- As for the 8 national nature reserves in Qinling Natural Reserve of China, the author should elaborate their geographical location and other information. Please try to draw a map to show them.
Response: We agree with the reviewer’s insightful opinion. A map has been added to show the geographical location and other information, please see the figure 2.
- The format of references in line 317 is incorrect, please correct.
Response: We agree with the reviewer’s comment, we have revised the format of references. Please see the revised paper.
- Please specify the writing format of CO2. 2 You need to set the subscript form, such as line 381 and Figure.2.
Response: According to the reviewer’s suggestion, we have specified the writing format of CO2, please see the the revised paper.
- Please correct the format of 383 rows and 384 rows. Table 5 should start a new row.
Response: According to the reviewer’s reminder, we correct the format of table, please see the update Table 5.
- Please standardize the format of Table 5 and try to keep some data in one row.
Response: According to the reviewer’s suggestion, we have standardized the format of Table 5 accordingly.
- More references need to be introduced in the discussion part. At present, there are few references in the whole discussion part, and some need to be cited to support key points. In addition, the author should try his best to compare the research results of Qinling Natural Reserve with those of other regions based on the research status at home and abroad.
Response: We agree with the reviewer’s suggestion of more references need to be introduced in the discussion part. A more detailed analysis and some references to the corresponding sentence are added in section of Disscussion part, especially the first paragraph in Disscussion as the follows:
“... This finding is consistent with those of Kuo & Chen [25], who found that transportantion sector consumes the largest energy (67%) and generate the largest proportion of CO2 emissions (68%) among different tourism sectors in Taiwan’s Penghu Islands region in China. According to Kuo et al. [25], the per day energy usage of an average domestic tourist in the Penhu Island is 1606 MJ and in the Kinmen Island is 1387MJ. This implies that tourists in Qingling Mountains in 2014, 2016 and 2019 use less energy than Penhu Island tourists and Kinmen Island tourists. However, protected areas have a policy priority for environmental protection, and thus they may be more likely to embrace sustainable development ideas than other “non-protected” areas [44]. Hence, local governments should propose effective trategies to encourage the inclusion of sustainable tourism ideas into protected area policies. The present study used the calculations of CO2 emissions per tourist per trip to examine the factors that might significantly influence the low-carbon behavior described in the literature. The regression results showed that low-carbon attitude, low-carbon knowledge, environmental education, policy reward, and income all strongly influence the low-carbon decisions made by tourists.”
- Please standardize the reference format.
Response: We appreciate the concern addressed by the reviewer. We have standardized the reference format, please see the the revised paper.

Reviewer 2 Report
The dual carbon targets and climate change strategies require industries to respond to low-carbon strategies and promote green development. With the development of tourism, the increase of the number of tourists and the change of consumption behavior, the carbon emission of tourism can not be ignored. This topic has certain practical significance. But there are still obvious deficiencies, which need to be improved by the author. The main manifestations are as follows:
1. Energy consumption includes the consumption of clean energy. Carbon emissions are mainly related to fossil energy consumption. In tourism consumption, the author needs to change the energy consumption into fossil energy consumption in the expression.
2. The expression of scientific issues and literature review in the introduction need to be further improved.
3.222 and 223 Lines: Statistical analysis of the basic characteristics of the questionnaire revealed that most tourists traveled by private vehicle to the protected areas of the Qinling Mountains. "This part describes the variable situation and does not need to describe the conclusion of the data.
4. Need to check whether there are any errors in the formula? For the part of Formula (1), the case of formula and text is inconsistent; In formula (2), the variables with explanatory parts are not used in italics; The case in formula (3) is inconsistent with the literal interpretation, and the meaning of each variable in formula (3) needs to be explained.
5. FIG. 2 and FIG. 2 are not standard and it is difficult to determine the corresponding relationship between chart variables and primary and secondary coordinate axes.
6. Figure 3 is not standard and the format needs to be optimized.
7. The English language expression of the full text still needs to be further improved.
Author Response
Measurement of tourism-related CO2 emission and the factors influencing low-carbon behavior of tourists: Evidence from protected areas in China
We appreciate the comments and suggestions from the editor and reviewers. We have sincerely considered all the comments and revise the paper accordingly. For the detailed corrections, please see the revised manuscript, in which we highlight the major revision in Red. We also provide this point-by-point response to all the questions raised by the reviewers, in which our responses are highlighted in Blue while the reviewers’ and editor’s comments are kept in black.
Reviewer #2
- Energy consumption includes the consumption of clean energy. Carbon emissions are mainly related to fossil energy consumption. In tourism consumption, the author needs to change the energy consumption into fossil energy consumption in the expression.
Response: According to the reviewer’s suggestion, we have rephrased the corresponding sentence, please see the the revised paper.
- The expression of scientific issues and literature review in the introduction need to be further improved.
Response: We agree with the reviewer’s comments. In the revised paper, we deepen the analysis in the Introduction part of the paper in order to emphasize the scientific issues. And some references to the corresponding sentence are added in section of Introduciton. Some added texts in the revised paper as the follows:
“…In the tourism context, protected areas has emerged as an important concept in recent outdoor recreation research [6]. In protected areas, tourism activity is usually restricted to areas where the environmental impact can be minimized, and the protected areas are often clearly delineated. As popular tourist destinations for leisure activities, protected areas help promote tourism for poverty alleviation and sustainable development [7]…. As the ongoing growth of tourism and rising concerns about its sustainability, it will become ever more important to understand the energy requirements and carbon emissions of tourism in protected areas and to guide tourists towards low-carbon behavior. For the protected areas of China, however, CO2 emissions and the factors influencing tourist low-carbon behavior have seldom been evaluated or investigated. For sustainable development of tourism in protected areas, policies designed to mitigate impacts of human behavior require accurate measurement of CO2 emissions associated with tourism activity [9].”
- 222 and 223 Lines: Statistical analysis of the basic characteristics of the questionnaire revealed that most tourists traveled by private vehicle to the protected areas of the Qinling Mountains. This part describes the variable situation and does not need to describe the conclusion of the data.
Response: We agree with the reviewer’s comment and have rephrased the corresponding sentence, please see the section of Statistical analysis.
- Need to check whether there are any errors in the formula? For the part of Formula (1), the case of formula and text is inconsistent; In formula (2), the variables with explanatory parts are not used in italics; The case in formula (3) is inconsistent with the literal interpretation, and the meaning of each variable in formula (3) needs to be explained.
Response: We agree with the reviewer’s comments and correct expression of formula and delete some unclear and repetitive parts. Please see the details in the revised paper.
- 2 and FIG. 2 are not standard and it is difficult to determine the corresponding relationship between chart variables and primary and secondary coordinate axes.
Response: As the reviewer’s suggestion, we have optimized the figure 2 accordingly, please see the figure 2 in the revised paper.
- Figure 3 is not standard, and the format needs to be optimized.
Response: As the reviewer’s suggestion, we have optimized this figure, please see the figure 3 in the revised paper.
- The English language expression of the full text still needs to be further improved
Response: We thank the reviewer’s careful reading to point out language errors. In the revised paper, we have double checked the language and corrected the grammatical errors.

Reviewer 3 Report
The paper is well-written, leading to very interesting results in the field of green travel and eco-tourism. Congratullations for your work!
It is not clear for me if the study has research limitations, as the authors haven't declared any.
Only minor corrections are needed, in my opinion, for instance:
Line 60: - fxed
Line 65 - It will become ever more important
Author Response
Measurement of tourism-related CO2 emission and the factors influencing low-carbon behavior of tourists: Evidence from protected areas in China
We appreciate the comments and suggestions from the editor and reviewers. We have sincerely considered all the comments and revise the paper accordingly. For the detailed corrections, please see the revised manuscript, in which we highlight the major revision in Red. We also provide this point-by-point response to all the questions raised by the reviewers, in which our responses are highlighted in Blue while the reviewers’ and editor’s comments are kept in black.
Reviewer #3
- Line 60: - fixed.
Response: We agree with the reviewer’s comment and have rephrased the corresponding sentence, please see the revised paper.
- Line 65 - It will become ever more important.
Response: According to the reviewer’s suggestion, we have revised the corresponding sentence. Please see the revised paper.

Round 2
Reviewer 2 Report
In Figure 2, it is better to add a suitable projection for the map of China. In addition, it is required to draw a nine-dash line involving China's maritime territory. It involves territorial scope, which must be strictly required in the cartographic specification of China. Very important! Please be sure to modify it!
Author Response
In Figure 2, it is better to add a suitable projection for the map of China. In addition, it is required to draw a nine-dash line involving China's maritime territory. It involves territorial scope, which must be strictly required in the cartographic specification of China. Very important! Please be sure to modify it!
Response: We agree with this key comment and the map is redrawn, please see the Figure 2. A nine-dash line involving China's maritime territory and a nine-dash line involving China's maritime territory are added in the revised Figure 2. The map of China uses an azimuthal equidistant projection which with a central longitude of 105° and a starting latitude of 35°.
